# Molecular Insights to the Structure-Interaction Relationships of Human Proton-Coupled Oligopeptide Transporters (PepTs)

**DOI:** 10.3390/pharmaceutics15102517

**Published:** 2023-10-23

**Authors:** Yining Luo, Jingchun Gao, Xukai Jiang, Ling Zhu, Qi Tony Zhou, Michael Murray, Jian Li, Fanfan Zhou

**Affiliations:** 1Molecular Drug Development Group, Sydney Pharmacy School, Faculty of Medicine and Health, The University of Sydney, Sydney 2006, Australia; yining.luo@sydney.edu.au (Y.L.); jgao2708@uni.sydney.edu.au (J.G.); michael.murray@sydney.edu.au (M.M.); 2National Glycoengineering Research Center, Shandong University, Qingdao 266237, China; xukai.jiang@sdu.edu.cn; 3Macular Research Group, Save Sight Institute, Faculty of Medicine and Health, The University of Sydney, Sydney 2006, Australia; ling.zhu@sydney.edu.au; 4Department of Industrial and Physical Pharmacy, College of Pharmacy, Purdue University, West Lafayette, IN 47907, USA; tonyzhou@purdue.edu; 5Biomedicine Discovery Institute, Department of Microbiology, Monash University, Melbourne 3800, Australia; jian.li@monash.edu

**Keywords:** proton-coupled oligopeptide transporters, structure-interaction relationships, drug optimisation, drug development

## Abstract

Human proton-coupled oligopeptide transporters (PepTs) are important membrane influx transporters that facilitate the cellular uptake of many drugs including ACE inhibitors and antibiotics. PepTs mediate the absorption of di- and tri-peptides from dietary proteins or gastrointestinal secretions, facilitate the reabsorption of peptide-bound amino acids in the kidney, and regulate neuropeptide homeostasis in extracellular fluids. PepT1 and PepT2 have been the most intensively investigated of all PepT isoforms. Modulating the interactions of PepTs and their drug substrates could influence treatment outcomes and adverse effects with certain therapies. In recent studies, topology models and protein structures of PepTs have been developed. The aim of this review was to summarise the current knowledge regarding structure-interaction relationships (SIRs) of PepTs and their substrates as well as the potential applications of this information in therapeutic optimisation and drug development. Such information may provide insights into the efficacy of PepT drug substrates in patients, mechanisms of drug–drug/food interactions and the potential role of PepTs targeting in drug design and development strategies.

## 1. Introduction

Oligopeptides are defined as peptides characterised by a relatively short chain of amino acids, typically comprising fewer than 20 amino acid residues, whereas polypeptides refer to those exceeding 20 amino acids [1].

Oligopeptide transporters (PepTs), also known as Proton-coupled Oligopeptide Transporters (POTs), are membrane proteins responsible for transporting various dipeptides, tripeptides and peptide-like drugs across biological membranes [2]. PepTs belong to the Solute Carrier Transporter (SLC) superfamily of membrane transporters that mediate the uptake of exogenous and endogenous substances.

There are four PepT isoforms in humans: PepT1 (*SLC15A1*), PepT2 (*SLC15A2*), PhT1 (*SLC15A4*) and PhT2 (*SLC15A3*). PepT1 and PepT2 are the most important and best studied isoforms that are extensively involved in drug transport. This review mainly focuses on the major PepT isoforms PepT1 and PepT2.

The earliest research on PepTs was conducted using brush border membrane vesicles (BBMVs) and found that peptides are transported via an active mechanism dependent on Na^+^ [3]. In a subsequent investigation, it was noted that renal tubular cells have a unique transport system for dipeptides, tripeptides and peptide-like drugs coupled with the H^+^ gradient [4,5]. Hediger et al. cloned the first peptide transporter in mammals (rabbit PepT1) and undertook a functional characterisation [6]. Molledo et al. screened twenty-eight peptides to assess their binding interactions with PepT_st_ from *Streptococcus thermophilus* [7]. The findings suggested that PepT_st_ is able to accommodate diverse peptide structures because amino acid residues and water molecules in the active site are able to move and because the position of the peptide itself can be adjusted to enhance the fit [7].

Such information suggested that the structure–function relationship of PepTs may be used to optimise the interaction between PepTs and its drug substrates, which could be employed in drug design. To date, PepT-related drug development strategies have not been extensively discussed. Therefore, this review summarises recent knowledge regarding the structure-interaction relationships (SIRs) of PepTs and their substrates and discusses the potential application of this information in therapeutic optimisation and drug development.

### 1.1. Substrate Specificities of PepT1 and PepT2

Both PepT1 and PepT2 have broad substrate specificities, although distinct differences in substrate binding affinities and capacities have been reported for the transporters [8].

PepT1 is a low-affinity and high-capacity transporter. The drug substrates of PepT1 include renin inhibitors [9], angiotensin-converting-enzyme (ACE) inhibitors [10,11], beta-lactam antibiotics [12], thrombin inhibitors [13], delta-aminolevulinic acid, acyclovir, ganciclovir, and valganciclovir [14,15,16,17]. While PepT1 was able to bind the majority of the peptides that were evaluated, individual amino acids and tetrapeptides were not substrates [18]. A number of structural and physicochemical features influence the binding of substrates to PepT1. It was demonstrated that free terminal carboxyl and amino functional groups and the nature of the side chain were important substrate features [18]. While investigating the intestinal transport of beta-lactam antibiotics, it was observed that L-isomers are selectively transported across the cell membrane [19]. Brandsch et al. found that although PepT1 can transport a wide range of substrates, the spatial arrangement within the region between the C_α_ atoms of the peptide is important [20]. The absorption of intestinal peptides, especially those containing tertiary amide bonds, is also influenced by cis/trans isomerisation because only the trans conformer is transported [20]. The N-terminus of PepTs is important for the binding of dipeptides and tripeptides and that a C-terminal histidine residue may also be critical for the binding of dipeptides to PepT1 [21]. It has been suggested that hydrophobicity and structural rigidity may influence substrate recognition by PepT1 [22,23,24,25,26].

In contrast to PepT1, PepT2 is a high-affinity and low-capacity transporter. PepT2 can transport 400 dipeptides and 8000 tripeptides consisting of 20 essential L-α-amino acids [27,28]. Apart from short peptides, PepT2 also has the capacity to transport an extensive range of peptide-like drugs, including beta-lactam antibiotics and ACE inhibitors [2,29]. Some information is available on the structural and chemical features that modulate the interaction of beta-lactam substrates with PepT2. Daniel et al. reported that not all beta-lactam antibiotics are substrates for renal PepT2 [30]. Only those possessing an alpha-amino group within the phenylacetamido moiety, such as amino cephalosporins, were found to be substrates [30]. The α-amino group is believed to interact with a histidine residue in PepT2, which contributes to substrate recognition. The presence of a free α-amino group appears to enhance substrate affinity but may not be essential [31,32]. The presence or absence of peptide bonds does not appear to affect substrate recognition by PepT2 [16]. Thus, the dipeptide anserine, which has a β-amino acid in its N-terminal, has a high affinity for PepT2, and the hydrophobicity of the N-terminal region of aminopenicillin increases its affinity for PepT2 [16,33]. A free N-terminus and optimal conformations of carbonyl and carboxylate groups within the rest of the structure appear to promote substrate recognition [34]. Furthermore, the presence of hydrophobic side chains in peptides may also influence substrate recognition by PepT2 [35]. For example, a large aromatic hydrophobic group at the N-terminus of dipeptides has been shown to increase binding to PepT2 [36].

### 1.2. Tissue Localisation of PepT1 and PepT2

The tissue localisation of transporters is closely linked to their physiological and pharmacological roles. PepTs are expressed mainly in the small intestine and kidney and at lower levels in several other tissues. A recent report demonstrated that PepT1 is highly expressed in the small intestine [37], where it mediates the absorption of dipeptides and tripeptides from dietary proteins and gastrointestinal secretions. PepT2 is expressed in the glial cells of the intestine [38].

In the kidney, PepT1 and PepT2 act synergistically to absorb peptide-bound amino acids. PepT1 is expressed in the anterior region of the proximal convoluted tubules, while PepT2 is present in the posterior region but not other regions of the nephron [39].

In the brain, PepT2 is expressed in the cerebral cortex, olfactory bulb, basal ganglia, cerebellum, and hindbrain slices, with the highest abundance detected in the cerebral cortex [40]. PepT2 facilitates the removal of neuropeptides, peptide fragments and peptide-like drugs from cerebrospinal fluid and regulates neuropeptide homeostasis in extracellular fluid in the brain.

Overall, PepT1 is primarily involved in substrate uptake by the small intestinal epithelium, while PepT2 is more important in renal uptake mechanisms.

### 1.3. Transport Mechanism of PepT1 and PepT2

The transport mechanisms of PepTs are similar. It has been proposed that dipeptide and tripeptide transport is coupled to proton transport. The transporter activity of PepTs depends on an electrochemical proton gradient, during which protons and Na^+^ are exchanged. The activity of the Na^+^-K^+^ ATPase enzyme determines the rate of ion exchange. Dipeptides and tripeptides are hydrolysed in the cytosol after entering enterocytes (Figure 1) or nephrons (Figure 2), and free amino acids are then transported into the bloodstream by different amino acid transporters located at the basolateral membrane.

A recent study demonstrated a role for Ca^2+^ in the transport mechanism of PepT1 [41]. After absorption, a dipeptide or tripeptide stimulates the calcium-sensing receptor (CaSR) that is located at the basolateral membrane of enterocytes, which activates phospholipase C (PLC) and increases [Ca^2+^] flow through intermediate conductance Ca^2+^-activated K^+^ channels (IKCa). Opening IKCa channels allows the efflux of K^+^, which controls cell hyperpolarisation, providing the driving force for transepithelial dipeptide/tripeptide uptake via PepT1. However, whether a similar Ca^2+^-dependent transport mechanism occurs with PepT2 has not been shown [41].

### 1.4. Molecular Regulation of PepT1 and PepT2

Both PepTs are regulated in physiological and diseased states. The expression and activity of PepT1 are regulated by epidermal growth factor (EGF) signalling [42], certain drugs (e.g., cephalexin) [43] and hormones (e.g., insulin) [44]. In EGF-treated human intestinal Caco-2 cells, the expression and transport activity of PepT1 are decreased [42]. It has been shown that calcium channel blockers can increase the activity of PepT1 by reducing the Ca^2+^ concentration and increasing Na^+^ ion concentrations [45,45].

Studies have revealed that some dietary conditions modulate PepT1 expression and activity in the gut [46,47]. For instance, fasting increased intestinal PepT1 expression in mice, which increased the oral absorption of glycylsarcosine (Gly-Sar); this did not occur in PepT1 knockout mice [48]. Fasting for 24 h induced PepT1 mRNA and protein expression in rats to levels around two-fold of control [49]. In addition, amino acid supplementation reduced PepT1 protein expression in the jejunal mucosa by 30% [50].

microRNAs (miRNAs) are short noncoding RNA sequences of 21 to 23 nucleotides that silence genes by binding to complementary sequences within the mRNA molecule. miRNAs play numerous roles in developmental timing, differentiation, immunity, cell migration and barrier function in the small and large intestines [51,52]. Dalmasso et al. found that miRNA-92b (miR-92b) inhibited the mRNA and protein expression and transport function of PepT1 in human C2BBe1 cells [53].

Janus kinase 2 (JAK2) and Janus kinase 3 (JAK3) are involved in the Janus kinase/signal transducer and activation of the transcription (JAK/STAT) signalling pathway, which is a central regulator of cell proliferation and apoptosis. Both kinases positively regulate PepT1 and PepT2 function and expression [54,55]. Protein Kinase C (PKC) activation has been found to inhibit PepT1-mediated Gly-Sar uptake in the human colon carcinoma Caco-2 cell line [56]. PKC altered the maximal velocity of transport rather than substrate binding. PKC did not alter PepT1 protein synthesis or its transmembrane pH gradient [56]. Another study that investigated PepT1-mediated intestinal nutrient absorption reported that the increased transport capacity of PepT1 in IL-10(-/-)mice treated with *Lactobacillus plantarum* could be due to PKC activation [57].

Protein–protein interactions have also been found to modulate PepTs function and expression. A functionally significant interaction between the PDZ (PSD95, D1g and ZO1) domain-containing protein PDZK1 and the C-terminus of PepT2 has been demonstrated [58]. Noshiro et al. verified such findings in HEK293 cells and demonstrated that PDZK1 increased the cell surface expression of PepT2 [59]. Moreover, subsequent findings have shown that genetic polymorphisms of PDZK1 may alter the transport function of PepT2 [60]. In addition, it has been shown that PDZK1 stimulates the transport activities of PepT1 and PepT2 in mice [61]. Apart from PDZK1, another PDZ-domain-containing protein—NHERF2 (Na^+^/H-exchanger regulatory factor)—was also found to modulate the PepT2-mediated transport of peptides and peptidomimetic drugs due to altered post-translational regulation [62].

### 1.5. Regulation of PepT1 and PepT2 in Disease

The dysregulation of PepTs may contribute to the pathogenesis of human diseases [63,64]. Inflammatory bowel disease (IBD) is a chronic condition characterised by generalised inflammation and intestinal injury [65]. Standard physiological patterns of the gut are altered in patients with IBD, and peptides that may trigger pro-inflammatory effects are present at low levels in the cytoplasm of colonic epithelial cells [66]. Although the underlying pathological mechanisms of IBD remain unclear, increasing evidence suggests that an enhanced activity of PepTs facilitates the transport of bacteria-produced dipeptides/tripeptides into enterocytes that could enhance the immune response and promote colonic inflammation [63,67,68,69].

A recent study reported that the expression of PepT1 is related to colitis-associated tumorigenesis in a mouse model of colorectal cancer [70]. Consistent with this finding, the expression of PepT1 was increased in human colon biopsies [70]. Altered colonic PepT1 expression during inflammation is likely associated with the dysregulation of specific miRNAs in colorectal tumours [71,72].

The expression of PepT1 is dysregulated in diabetes [44,73,74,75]. Studies conducted in rodents have suggested that the tissue distribution, protein expression and/or transport activity of PepT1 and PepT2 may be altered due to aging and hypertension [64,76,77]. These findings have suggested that nutrient regulation and drug response (e.g., to ACE inhibitors) may be altered in populations with underlying medical conditions.

### 1.6. Genetic Polymorphisms of PepT1 and PepT2

To date, several coding polymorphisms and >100 haplotypes have been identified in PepT1 that is encoded by the *SLC15A1* gene [78,79]. Several PepT variants exhibit altered transport function.

Dipeptide substrate uptake mediated via eight nonsynonymous PepT1 variants was assessed by Zhang et al. [79]. The PepT1-F28Y variant displayed markedly decreased cephalexin and Gly-Sar uptake, which may be due to impaired substrate binding affinity (Table 1) [79,80]. Zhang et al. characterised nine PepT1 variants, all of which retained pH dependence and binding affinity for Gly-Sar and cephalexin [69]. Interestingly, the decreased activity of the P568L variant transporter has been attributed to an impaired turn-over rate [79]. S117N and G419A are the two most common PepT1 variants, which have preserved transport activity for a variety of substrates including Gly-Sar, enalapril, 5-aminolevulinic acid hydrochloride, captopril, cefadroxil, L-DOPA, cephalexin, and bestatin [78,79,81].

## 2. Structure–Function Relationships of PepT1 and PepT2

Recent studies have attempted to resolve the topology and structure of PepT proteins to provide a foundation for the development of SIRs that might assist the understanding of their roles in drug disposition.

### 2.1. Topology Model of PepTs

Wang et al. suggested that PepT1 and PepT2 show qualitatively similar patterns of molecular evolution, which is consistent with structural conservation between PepTs [28]. The hydrophobicity of the transmembrane domains (TMDs) in PepT1 and PepT2 is similar [28]. The topology model predicts that PepTs have 12 TMDs and an extracellular loop between TMD 9 and 10 [80,87]. However, this loop may not be essential for transporter function [86]. The N-terminus and TMDs are thought to adopt a pore-like structure, while TMD 7 to 9 may form the core of the substrate binding pocket [80]. In the case of PepT1, the first six TMDs constitute the central region of the substrate binding pocket and may have a role in pH dependence [88]. It has been suggested that residues 1 to 59, which span TMD1 and extracellular loop 1, and which extend into TMD2, may interact with the side chains of dipeptides, while residues 60 to 91 control the pH dependence of PepTs [89]. Mutagenesis studies have demonstrated that H57 in TMD 2 of PepT1 may be associated with proton binding while the two adjacent tyrosine residues (Y56 and Y64) may stabilise the proton charge (Table 1) [83]. H121 in TMD 4 of PepT1 could be involved in substrate recognition by promoting the charge neutralisation of acidic peptides prior to translocation [83,84]. Moreover, the increased affinity of the Y56F PepT1 variant for its substrates may be due to its ability to stabilise the H57 residue as the neutral form [86].

In PepT1 W294, E595 and Y167 are important residues for the recognition of specific substrates [82,85]. W294, as well as E26 and Y588, are involved in the initial binding interaction with substrates, while E595 and Y167, as well as Y12, R282 and D341, regulate substrate translocation [82]. Examination of the distances between particular residues in PepTs can be considered in relation to intermolecular distances between substrate atoms, which may offer insight into drug design approaches [86].

In the case of PepT2 (Figure 3), the residues R57, H87 and H142 have been shown to be critical for substrate binding and transport activity (Table 2) [84,87]. Furthermore, the corresponding histidine residues (H57 in PepT1 and H87 in PepT2) were found to adopt similar topological locations, which allows them to participate in H+ binding and substrate translocation [86]. TMDs 2 and 3 likely contribute to pH dependence, while TMDs 9 to 10 are critical for functional divergence in the hydrophobic regions of PepT2 [90]. The early study also indicated that the N-terminus of PepT2 is critical for its phenotypical characteristics such as the selectivity and affinity for substrates, pH dependence and electrophysiological properties [91]. The three amino acid and tyrosine-based motifs at the C-terminus of PepT2 are critical to membrane localisation [92].

N-glycosylation is a unique post-translational modification in eukaryotes that influences protein function by modifying specific asparagine residues with oligosaccharides [93]. Many studies have showed that N-glycosylation is essential for the localisation, stability, and substrate binding of SLC transporters [94,95]. N-glycosylation of PepT1 accounts for around one-third of its total mass and shows diverse patterns of oligosaccharide structures in different tissues [96]. N-glycosylation is crucial for maintaining PepT1 transport activity by preserving cell surface protein expression [97]. N-glycosylation may also contribute to the resistance of PepT1 to proteolytic cleavage by proteinase K and retain its intrinsic stability against trypsin [98].

### 2.2. D Structure of PepT1 and PepT2

Crystal structures for mammalian membrane proteins are difficult to obtain due to protein size and the intricacies of intra- and extracellular loops [86].

Three-dimensional (3D) structures of several bacterial PepT homologues have recently been resolved, although those of human analogues remain largely unknown. Resolution of the structure of PepT_so_ from *Shewanella oneidensis* [99] was followed by the structures of PepT_St_ from *Streptococcus thermophilus* [100], GkPOT from *Geobacillus kaustophilus* [101], PepT_so2_ from *Shewanella oneidensis* [102], YePEPT from *Yersinia enterocolitica* [103,104], PepT_Xc_ from *Xanthomonas campestris* [105] and rPepT2 from rat [106]. There are several common structural features shared by these PepT analogues: they all contain the canonical major facilitator superfamily (MFS) fold with 12 transmembrane helices; the N- and C-terminal helix-bundles are formed, and these PepTs may adopt a “V” shaped conformation. However, the functional significance of these signature structural features remains largely unknown.

It has been predicted that the open and closed states of PepTs are controlled by hinge-like movements at the apex of the H10–H11 regions [100], so that the binding of both dipeptides and the larger tripeptides may be accommodated [107]. On the other hand, however, a recent study suggested that di- and tripeptides bound similarly to PepTs [108]. The potential flexibility of the binding pocket in PepT2 may be a major hurdle for the development of structure-based drug design.

Studies with the bacterial PepT homologue GkPOT are consistent with H^+^-coupled peptide symport for PepTs [101]. When binding to the carboxyl groups of peptide substrates, the deprotonation of E310 may promote intracellular substrate release. A salt bridge between the E310 and R43 residues may facilitate conformational change. Another study revealed that proton-bound PepTs extracellularly redirect the inward- to outward-facing status [105]. The previous study has also explored the role of extracellular domains (ECDs) of PepTs [109]. The interaction between trypsin and ECDs likely improves the substrate uptake efficiency of PepTs.

A cryogenic electron microscopic structure of human PepT1 and PepT2 was not fully consistent with earlier findings regarding transporter substrate recognition and movement [110]. As predicted by topology models, PepTs have 12 TM helices and a long linker region that connects both helical bundles. However, the unique architecture of PepTs including the last 25 residues at the C-terminus and the first 40 residues at the N-terminus of PepT2 could not be determined largely due to the low abundance of PepTs purified [110]. In reference to the bacterial homologues, the N-bundle of PepT1 and PepT2 shows greater flexibility and dynamics, while the C-bundle shows greater rigidity. This may be due to the additional extracellular region between TM9 and TM10 in PepTs requiring the c-bundle to stabilise such a structure [102,107,111].

The mechanism of substrate transport in PepTs has been proposed from cryogenic electron microscopy structures. PepTs are initially in an outward state facing the extracellular side; this is stabilised by two salt bridges. During substrate binding, substrates are accommodated in the charged central cavity of the PepT, which causes the bending of the N-bundle that in turn makes the central cavity assembly tighter. The transporter protein then adopts an inward-facing state that is stabilised by a single salt bridge. The movement of TM4 and TM5 away from TM10 and TM11 exposes the cytosolic side and promotes substrate release into the cytoplasm [107].

## 3. Structure-Based Drug Design and Optimisation in Relation to PepTs

Understanding SIR models of membrane transporters and their substrates may form the basis of the design and development of drugs that rely on transporters to move across cell membranes. Such transporter-related strategies include (1) the manipulation of the drug structure to modulate their interactions with transporters, (2) the development of novel, customised transporter-targeted drug delivery carriers, and (3) the development of an individualised regimen based on transporter pharmacogenomics.

The design of drugs that interact favourably with specific transporters may have advantages for the delivery of drugs to target organs, which could improve bioavailability and overcome challenges. Recently, Foley et al. conjugated a range of marketed drugs with modified dipeptides and assessed the affinities of their binding with PepT1 in oocytes and Caco-2 cells [112]. The prodrugs that interacted favourably with PepT1 were further evaluated in rats to test whether such a PepT1-targeted approach can improve the oral permeability of clinically used drugs [112]. It has also been reported that the conjugation of specific dipeptides with nanoparticle drug carriers may improve the oral delivery of drugs like docetaxel [113] and cyclosporine A [114]. Such peptide-conjugated nanoparticles showed enhanced affinity for PepT1 and improved oral bioavailability. In addition, PepT1-mediated prodrug design has been shown to greatly increase the affinity of PepT1 for cyclic dipeptides [115] and drugs such as peramivir [116], which improves their oral bioavailability. Interestingly, 5-aminosalicylic acid (5-ASA) conjugated to amino acids have been shown to be substrates to PepT1 [117]. 5-ASA is a front-line agent for IBD, but its delivery to inflamed colonic sites is poor. Thus, conjugated 5-ASA derivatives may represent a new strategy to improve treatment outcome in patients with IBD. Together, these findings suggest that PepTs may serve as potential targets that could improve the bioavailability of orally delivered drugs.

Taking advantage of computer docking and molecular modelling, the recent study of Khavinson et al. predicted the binding of PepT1 and two L-amino acid transporters (LATs) with >8000 di-, tri-, and tetra-peptides [118]. There were 26 ultrashort peptides that were identified that exhibited relatively higher binding affinities for PepT1 and LATs. Although the biological activities of these ultrashort peptides are not fully understood, it is plausible that their interactions with PepTs and LATs may contribute to their pharmacological effects, such as anti-cancer actions.

Although further optimisation and clinical testing are required, the present findings suggest that SIR models of PepTs in drug design may be useful in the further therapeutic development [119].

Information on the structural biology of PepTs contributes to delineating their interactions with substrates. The acetylated form of proline–glycine–proline (Ac-PGP) is a collagen-derived matrikine that can be found in the lungs of patients with chronic inflammatory disease. In a mouse model of acute LPS-induced lung injury, PepT2 knockout increases the level of Ac-PGP and inflammation in bronchoalveolar lavage and lung tissue, which suggested that PepT2 may play an important role in redistributing the endogenous bioactive peptides in acute respiratory distress syndrome (ARDS) [120]. Because ARDS is associated with viral pneumonia [121] and neutrophilia in COVID-19 [122], PepT2 may be a potential therapeutic target to minimise the side effects of COVID-19-related ARDS.

Polymorphisms of PepTs may alter the function and/or expression of the encoded transporters and thus influence the pharmacokinetics of drug substrates [123]. Liu et al. reported that the renal clearance of cephalexin differs in human subjects carrying different PepT2 polymorphisms [124]. Indeed, pharmacogenetic influences on drug pharmacokinetics and/or toxicities have been widely reported for several SLCs. For example, in the study of Zhou et al., specific genetic variants of organic anion transporting polypeptide 1A2 (OATP1A2) were associated with impaired transporter function and expression; this mechanism may contribute to interindividual differences in the response to imatinib [125].

Genetic variants of transporter proteins may alter the pharmacokinetics and clinical outcomes of drugs. It is important to obtain molecular insights into how such variants influence transporter function and expression. Understanding the relationship between the chemical structure of substrates and how pharmacogenetic variation influences the transporter function may provide insight into drug efficacy in different populations.

At present, the application of the above-mentioned strategies in relation to PepT targeting remains limited, especially regarding how the PepT structure influences substrate binding. Future research may involve the application of computational modelling and molecular dynamic simulations to resolve the SIR models of PepTs. Advances in this area could facilitate the development of novel therapeutics that utilise structural information on PepTs.

## 4. Conclusions

PepT1 and PepT2 are important influx transporters that mediate the cellular uptake of dipeptides, tripeptides, and peptide-like drugs. They have a profound role in drug disposition as well as maintaining homeostasis. Insights gained regarding the SIR models of PepTs could shed light on the design of new drugs or the optimisation of existing therapeutics in relation to their interactions with PepTs.

## Figures and Tables

**Figure 1 pharmaceutics-15-02517-f001:**
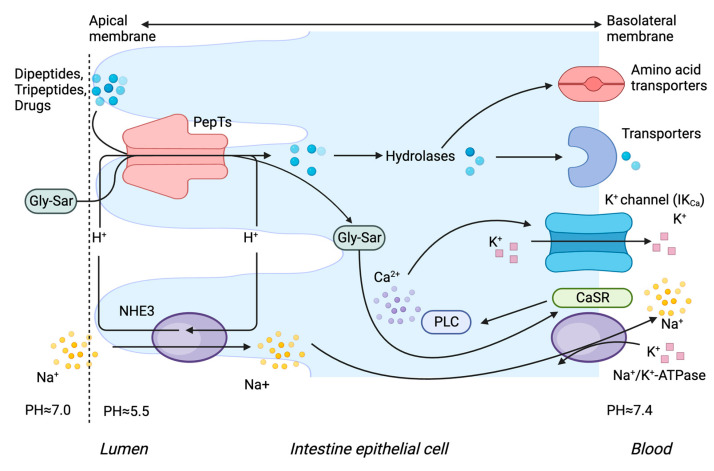
Transport mechanisms of PepT1 and PepT2 in intestinal epithelial cells. Key: CaSR: calcium sensing receptor; Gly-Sar: glycylsarcosine; PLC: phospholipase C; NHE3: Na^+^/H^+^ exchanger 3.

**Figure 2 pharmaceutics-15-02517-f002:**
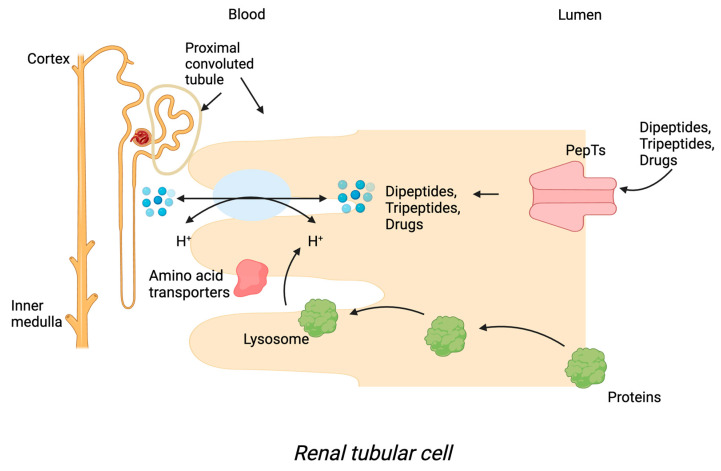
Transport mechanisms of PepT1 and PepT2 in renal tubular cells.

**Figure 3 pharmaceutics-15-02517-f003:**
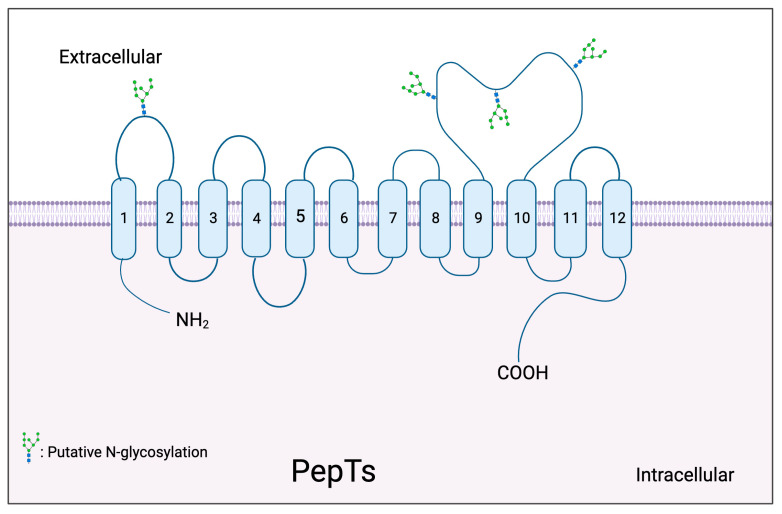
Predicted topology model of PepTs. Numbers in the figure indicate the predicted 12 transmembrane domains.

**Table 1 pharmaceutics-15-02517-t001:** Functional characterisation of PepT1 mutants.

Mutant	Putative Location	Transport Function	References
Y12A	TMD1	Gly-Sar uptake -	[82]
F28Y	Extracellular Loop 1	Cephalexin uptake ↓	[78]
Y56A	TMD2	Gly-Sar uptake -	[83]
Y56F	TMD2	Gly-Sar uptake ↓	[83]
H57N	TMD2	Gly-Sar uptake ↓↓	[84]
H57Q	TMD2	Gly-Sar uptake ↓↓	[84]
H57R	TMD2	Gly-Sar uptake -	[83]
Y64A	TMD2	Gly-Sar uptake -	[83]
Y64F	TMD2	Gly-Sar uptake ↓	[83]
H111C	Extracellular Loop 2	Gly-Sar uptake -	[83]
H111R	Extracellular Loop 2	Gly-Sar uptake -	[83]
S117N	Extracellular Loop 2	Gly-Sar uptake -	[81]
H121C	TMD4	Gly-Sar uptake ↓	[83]
H121N	TMD4	Gly-Sar uptake -	[84]
H121Q	TMD4	Gly-Sar uptake -	[84]
H121R	TMD4	Gly-Sar uptake ↓	[83]
Y167A	TMD5	Gly-Sar uptake ↓↓	[85]
Y167F	TMD5	Gly-Sar uptake ↓↓	[85]
Y167H	TMD5	Gly-Sar uptake ↓↓	[85]
Y167S	TMD5	Gly-Sar uptake ↓↓	[85]
H260N	Intracellular Loop 3	Gly-Sar uptake -	[84]
H260Q	Intracellular Loop 3	Gly-Sar uptake -	[84]
R282A	TMD7	Gly-Sar uptake -	[82]
W294A	TMD7	Gly-Sar uptake ↓↓	[82,86]
G419A	Extracellular Loop 5	Gly-Sar uptake-	[81]
P586L	TMD10	Glycyl-sarcosine uptake ↓	[78]
E595A	TMD10	Gly-Sar uptake ↓↓	[82]

Key: ↓: significantly reduced; ↓↓: almost abolished; -: unchanged.

**Table 2 pharmaceutics-15-02517-t002:** Functional characterisation of PepT2 mutants.

Mutant	Putative Location	Transport Function	Reference
R57H	TMD1	Gly-Sar uptake ↓↓	[87]
H87N	TMD2	Gly-Sar uptake ↓↓	[84]
H142N	TMD4	Gly-Sar uptake ↓	[84]
P409S	TMD9	Gly-Sar uptake -	[87]

Key: ↓: significantly reduced; ↓↓: almost abolished; -: unchanged.

## Data Availability

Not applicable.

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
