# Peer review of "Molecular Insights to the Structure-Interaction Relationships of Human Proton-Coupled Oligopeptide Transporters (PepTs)"

_pharmaceutics, 2023, doi:10.3390/pharmaceutics15102517_

Round 1
Reviewer 1 Report
Luo and colleagues summarized the structure-interaction relationships of human proton-coupled oligopeptide transporters (PepTs). The manuscript was generally well-written and well-organized. I have some minor comments.
1) Please clearly state the aim of this narrative review in the Abstract and Introduction.
2) Figure 1: Please correct the action of Na+/K+-ATPase. I guess potassium ions are incorporated into the cells, not ATP.
3) Figure 2: Amino acids are metabolized to dipeptides/tripeptides by cytosolic peptidases? I guess this is incorrect.
4) The abbreviation should be used after the definition of them. For example, JAK, STATA (line 163), SIR (line 218), and TM (280).
5) Please change the term "IBD patients" to "patients with IBD".
6) Please correct the typo "incfreased" (line 194).
7) Please use the abbreviation of Gly-Sar instead of glycylsarcosine in line 214.
8) Lines 194-195: Cited paper was the study using a murine model. Please correct the citing paper.
9) Lines 225-226: Please recheck the sentence. I can not understand.
10) Line 253: Please omit the term "high".
Reviewer 2 Report
The role of Pept in the transport of di-, tripeptides and peptide drugs into cell is an urgent topic of modern molecular medicine and pharmacology. The authors gave very informative figure 1 and 2. However, in our opinion, the article does not have sufficient scientific novelty and cannot be published in the Pharmaceuticals Journal.
The review analyzes 116 literary references. However, there are only 5 articles among them over the past 5 years (2019-2023). Thus, the review does not have sufficient scientific novelty, since it does not take into account a large number of publications over the past 5 years on the functions and Pept mechanisms of action. Examples of links to recent articles on Pept transporters can be found in our latest reviews on this topic for 2022 and 2023 https://www.mdpi.com/1422-0067/23/14/7733/htm and https://www.mdpi.com/2218-273X/13/3/552.
Line 83-85. Authors write that Pept2 can carry more than 400 dipeptides and more than 8000 tripeptides consisting of 20 essential amino acids. This is not true, since there are only 400 variants of dipeptides and only 8000 variants of tripeptides consisting of 20 essential amino acids. The word "more" should be deleted.
Reviewer 3 Report
The review gives an interesting overview of the interaction between PepTs and their substrates.
Minors
1) Lines 67: Please make "Delta-" lowercase.
2) Lines 37-38: Please indicate the reference.
3) Line 107: PepT1 may also be expressed in jejunum, as well as duodenum and ileum. Considering the title of this manuscript, human data should be cited, although ref#34 shows mouse PepT1 expressed along the small intestine.
4) Lines 123, 138, and 148: The terms "Na+ ions" and "K+ ions" may be those without "ions" or "+."
5) Fig.1: K ions released from the cytoplasm by IKCa should be taken up by Na+/K+-ATPase.
6) Line 138: IKCa channels mediates “the efflux” of K ions, which is involved in hyperpolarization.
7) As lines 108-109 describe that PepT2 is expressed in intestinal glial cells, PepT2 should be deleted from Fig.1, which shows intestinal epithelial cells.
8) Lines 149 and 225: Please check the grammar of the sentence.
9) Line 162: Caco-2-bbe should be C2BBe1.
10) Line 194: There is a typo.
11) Lines 197-202: The context of this paragraph is unclear. The inhibitory effect of glibenclamide on PepTs transport activity may not explain the insulin-regulated expression.
12) Line 246: “Fig. 3” should appear in the previous paragraph.
13) Line 279, H1 to H12 require explanation.
14) Lines 346-352: The example of transmembrane protein such as SLC and ABC transports is easier to understand the context than that of transthyretine, which is a soluble protein.
Reviewer 4 Report
The submission by Luo et al. is a neatly written review with a wealth of information about PepTs. The manuscript cites 116 papers, which covers a significant portion of available literature considering the relatively small overall number of papers published about these transporters. The paper is clearly written in perfect English, and provides the reader with a concise yet comprehensive overview of current knowledge. My comments are as follows:
Major
- I am unhappy with Figures 1 and 2 for a number of reasons. Firstly, the figures have most likely been adapted from different sources because they use different symbols for the same thing. E.g., PepTs themselves are depicted differently, and PepT substrates appear as blue balls in Figure 1 but have no graphical symbol in Figure 2. This is acceptable, i.e., does not interfere with understanding, but certainly not elegant. Secondly, both figures employ ambiguous or inadequate graphical design elements. In Figure 1, there is a second membrane-like undefined layer underneath the microvillous apical plasma membrane, which is challenging to interpret. In Figure 2, the schematic cell on the right is supposed to represent a renal proximal tubule epithelial cell but looks more like an amoeba. I would suggest the authors to improve and better harmonize Figures 1 and 2, which may involve some re-designing.
- Whereas the Introduction (Section 1) and Structure-function relationships (Section 2) cover the expectation raised by each section title, we learn relatively little about structure-based drug design and optimization from Section 3. The only citation that aligns well with this topic is the one about the new antituberculotic targeting MmpL3. Other papers cited in this section are only loosely related to PepT-centered rational drug design. In particular, I am struggling to see the logical link between destabilizing transthyretin mutations, polymorphic regions in MATP, and PepTs. The authors should either find more relevant citations (and literature is, I appreciate, scarce) or better clarify the unifying concept behind these seemingly unrelated topics.
Minor
- Line 194: “incfreased" is a typo
- Line 194-195 says “PepT1 was increased in human colon biopsies” – With colitis? With tumor?
- Lines 207-208: “Dipeptide substrate uptake […] was assessed” – should add “by Zhang et al.”
- Line 218: “SIRs” – please expand abbreviation at the first use
- Line 225: “How” should read “However, “
- Whole paragraph, lines 339-344: Apart from the above-mentioned fact that the cited findings have little to do with structure-based drug design, this paragraph is slightly confusing. It opens with the statement that cefadroxil was increased in the brain ECF and CSF of PepT2 KO mice, and concludes with the suggestion that PepT2 drug substrates may not achieve high levels in ECF or CSF. Physiologically? Or in the KO (where they do)? If the message is that PepT2 under physiological conditions helps keep its substrates out of brain ECF and CSF, the wording should be clearer.
Round 2
Reviewer 2 Report
Authors added 16 new references and discussed it in the review. However, this does not change the essence of the comment which I made on the essence of the article. As I have already written, most of 116 articles previously analyzed by the authors were published more than 10 years ago. Text and conclusion of the review do not contain novelty. Quite a lot of reviews analyzing modern literature have been published on this topic in the last 5 years. In this regard, I do not see any novelty in the article and do not recommend it for publication.
Author Response
Please see our response to reviewer 2 comments in the attachment.
